# Diabetes Mellitus in Prader-Willi Syndrome: Natural History during the Transition from Childhood to Adulthood in a Cohort of 39 Patients

**DOI:** 10.3390/jcm10225310

**Published:** 2021-11-15

**Authors:** Alice Clerc, Muriel Coupaye, Héléna Mosbah, Graziella Pinto, Virginie Laurier, Fabien Mourre, Christine Merrien, Gwenaëlle Diene, Christine Poitou, Maithé Tauber

**Affiliations:** 1Centre de Référence Maladies Rares (PRADORT, Syndrome de Prader-Willi et Autres Formes Rares d’Obésité avec Troubles du Comportement Alimentaire), Service d’Endocrinologie, Obésités, Maladies Osseuses, Génétique et Gynécologie Médicale, Hôpital des Enfants, 31059 Toulouse, France; alice.clerc@universite-paris-saclay.fr (A.C.); diene.g@chu-toulouse.fr (G.D.); 2Assistance Publique-Hôpitaux de Paris, Centre de Référence Maladies Rares (PRADORT, Syndrome de Prader-Willi et Autres Formes Rares d’Obésité avec Troubles du Comportement Alimentaire), Service de Nutrition, Hôpital Pitié-Salpêtrière, 75013 Paris, France; muriel.coupaye@aphp.fr (M.C.); helena.mosbah@aphp.fr (H.M.); christine.poitou-bernert@aphp.fr (C.P.); 3Assistance Publique-Hôpitaux de Paris, Service d’Endocrinologie, Gynécologie et Diabétologie Pédiatrique, Hôpital Necker-Enfants Malades, 75743 Paris, France; graziella.pinto@aphp.fr; 4Assistance Publique-Hôpitaux de Paris, Centre de Référence Maladies Rares (PRADORT, Syndrome de Prader-Willi et Autres Formes Rares d’Obésité avec Troubles du Comportement Alimentaire), Hôpital Marin d’Hendaye, 64701 Hendaye, France; virginie.laurier@aphp.fr (V.L.); fabien.mourre@aphp.fr (F.M.); christine.merrien@aphp.fr (C.M.); 5Inserm UMR 1295—CERPOP (Centre d’Epidémiologie et de Recherche en Santé des POPulations), Équipe SPHERE (Santé Périnatale, Pédiatrique et des Adolescents: Approche Épidémiologique et Évaluative), Université Toulouse III Paul Sabatier, 31062 Toulouse, France; 6UMRS 1269, Faculté de Médecine Sorbonne Université, INSERM, Nutrition et Obésité: Approches Systémiques «NutriOmics», 75006 Paris, France; 7Institut Toulousain des Maladies Infectieuses et Inflammatoires (Infinity) INSERM UMR1291—CNRS UMR5051—Université Toulouse III, 31062 Toulouse, France

**Keywords:** Prader-Willi syndrome, type 2 diabetes mellitus, syndromic obesity

## Abstract

Type 2 diabetes mellitus (T2DM) affects 20% of patients with Prader-Willi syndrome (PWS), with many cases diagnosed during the transition period. Our aim was to describe the natural history of T2DM in patients with PWS before the age of 25 years and to develop screening and preventive strategies. Thirty-nine patients followed in the French PWS Reference Center were included (median age 25.6 years [23.7; 31.7]). Twenty-one had been treated with growth hormone (GH), fifteen had not, and three had an unknown status. The median age at T2DM diagnosis was 16.8 years (11–24) and the median BMI was 39 kg/m^2^ [34.6; 45], with 34/35 patients living with obesity. The patients displayed frequent psychiatric (48.3% hospitalization,) and metabolic (56.4% hypertriglyceridemia,) comorbidities and a parental history of T2DM (35.7%) or overweight (53.6%) compared to the PWS general population. There was no difference in BMI and metabolic complications between the GH-treated and non-GH-treated groups at T2DM diagnosis. Patients with PWS who develop early T2DM have severe obesity, a high frequency of psychiatric and metabolic disorders, and a family history of T2DM and overweight. These results underline the need for early identification of patients at risk, prevention of obesity, and repeated blood glucose monitoring during the transition period.

## 1. Introduction

Prader-Willi syndrome (PWS) is a rare genetic disorder with an incidence of 1 in 21,000 births [1]. It is usually caused by the loss of paternally inherited imprinted genes at 15q11.2-q13. In about 60–65% of cases, the cause is a paternal 15q11-q13 deletion (DEL) and, in the remaining cases, the cause is a maternal uniparental disomy of chromosome 15 (mUDP; 25–30%) or, more rarely, an imprinting defect or translocation. It is now acknowledged that the entire phenotype is linked to hypothalamic dysfunction [2]. PWS is characterized by a unique developmental, nutritional, endocrine, metabolic and behavioral trajectory over a lifetime. In the neonatal period, infants display severe hypotonia and sucking-swallowing deficits that interfere with feeding and decrease weight gain. Then, they develop excessive weight gain, hyperphagia, and obsession towards food that leads to severe early obesity in the absence of strict control of food access in a firm and caring environment [3]. Obesity in PWS is multifactorial in origin. In addition to the eating disorder, patients have lower muscle mass and greater fat mass than in the general population with comparable BMI [4,5]; they also have lower resting energy expenditure [6,7]. The most common pituitary hormone deficits are growth hormone (GH) deficiency (80% of patients), hypogonadism of mixed origin (90% of patients), and central hypothyroidism (20–40% of patients) [8,9,10]. A few cases of adrenal insufficiency have also been described, but the prevalence appears to be low [11,12]. Since obtaining marketing authorization (MA) in 2000, GH supplementation has been systematic in children with PWS, with treatment starting in the first year of life. Its positive effects on adult height, body composition, BMI and metabolic status have been demonstrated in several studies [13,14,15]. Most patients have intellectual disability and behavioral disorders [16,17]. A study of 150 adults with PWS described a typical psychopathological pattern of the disease, emphasizing the following types: basic, impulsive, compulsive, and psychotic [18]. More recently, classifications of symptoms according to the Research Domain Criteria (RDoC) [19] and addiction models have been proposed [20].

T2DM is common in PWS and affects 20% of adults compared to 5–7% of the general population [21]. In France, 5% of adults were treated for diabetes in 2016 and, according to a health insurance survey, 92% of these cases were T2DM [22]. The exact T2DM prevalence among patients under 25 living with common obesity is unknown. A recent American study of 635 young people between 10 and 20 years with a BMI >85th percentile found that 31.5% were prediabetic and 6.1% had T2DM [23]. In Europe, T2DM affects about 5% of children over the age of 10 living with obesity. A 2006 French study of 308 overweight children (78.9% with obesity) from 7 to 17 years found that 42.5% showed insulin resistance and 3.6% showed glucose tolerance disorders [24]. The mean age at T2DM diagnosis in PWS is 20 years [21,25], and the data in the literature indicate a prevalence of less than 2% in patients under 18 [26]. In 2010, we reported that 4% of a cohort of 142 children with PWS showed glucose intolerance and no T2DM [8].

Two other studies in Japan and Korea have reported higher incidences and more pediatric cases [27,28]. BMI, the HOMA index and age were recently identified as risk factors in an Italian cohort study of 274 patients with PWS [29]. However, several studies suggest that the pathophysiological mechanisms behind T2DM in PWS may be different from those involved in common obesity. At an equivalent BMI, higher insulin sensitivity was observed in PWS subjects compared to subjects with common obesity [30]. Similarly, at comparable leptin levels (i.e., equivalent amounts of fat mass), children with PWS had a better response to exogenous insulin than those with common obesity [31]. This lower insulin resistance in PWS might be explained by the lower amount of visceral fat, the GH deficiency, and the increased levels of ghrelin and adiponectin [25]. Some studies also suggest pancreatic beta cell dysfunction; Schuster et al. observed decreased insulin secretion in PWS patients compared to patients with common obesity after oral glucose ingestion or IV glucose injection [32]. A defect in the pro-convertase 1 (PC-1) enzyme has been demonstrated in hypothalamic neurons obtained after redifferentiation of pluripotent stem cells recovered from skin biopsies. This would explain the high proinsulin levels and lower insulin levels found in patients with PWS [33]. In addition, high levels of acylated ghrelin (the “hunger hormone”), very frequently found in patients with PWS, might promote weight gain and the development of diabetes [34,35]. Moreover, the oxytocin dysfunction observed in PWS could explain a greater appetite for sugar. Murine and bovine studies have demonstrated the stimulatory role of oxytocin in glucose uptake by muscle cells and adipocytes, as well as in insulin secretion by pancreatic islets [36]. However, the observations made in humans do not yet allow conclusions to be drawn.

Despite possible differences in the pathophysiology of T2DM in PWS, the treatment recommendations are the same as for patients with T2DM in the general population. Healthy lifestyles that include dietetic guidelines are at the forefront of management, but they are difficult to achieve in patients with PWS—especially in certain families and during adolescence. They are almost always associated with pharmacological treatment, with metformin being the first-line therapy. When monotherapy is insufficient, sulfonylurea, glucagon-like peptide-1 (GLP-1) agonists (liraglutide, exenatide, semaglutide, dulaglutide) or dipeptidyl peptidase-4 (DPP-4) inhibitors may be combined. More recently, sodium glucose cotransporter-2 (SGLT-2) inhibitors have been used in patients with PWS in uncontrolled situations [37,38]. In the event of failure or insufficient efficacy of combination therapy, insulin treatment is initiated [25,39]. For patients under 18 in France, only insulin and metformin (for those over 10) are approved. Liraglutide has recently been approved in the United States for children over 10.

In this study, we sought to describe the characteristics of T2DM in adolescents and young adults with PWS and to identify the associated factors for directing the early detection and prevention of T2DM.

## 2. Materials and Methods

We conducted a multicenter retrospective descriptive study. All included patients were followed in big centers of the Prader-Willi Reference Center network. Patients or their parents had authorized the collection of information concerning them in the Center’s database. In this context, they provided no objection to the use of their data.

### 2.1. Patients and Inclusion Criteria

We decided to conduct our study in four big centers of the French Reference Center of the PWS network (Toulouse, Hendaye and Pitié-Salpétrière and Necker Hospital in Paris). These four centers regularly follow 342 adults and 305 children with PWS, thus comprising more than half of the patients followed in France. They have been working together since 2004 and share the same practices.

The inclusion criteria were the following: genetically confirmed diagnosis of PWS, followed in one of the four centers mentioned above, and diagnosis of T2DM before the age of 25 years.

There was no exclusion criterion. Thirty-nine patients were included.

### 2.2. Measurements and Data Collection

Data collection was carried out between July 2020 and April 2021. Clinical and biological data were recorded in the medical records as part of routine care.

We collected clinical and anamnestic data concerning PWS, T2DM (age, BMI and biological parameters at diagnosis, parental history, antidiabetic medications) and the transition (pediatric or non-pediatric follow-up, transition age if applicable).

Overweight was defined as BMI > 25 kg/m^2^. Obesity was defined as follows, according to the standards of the World Health Organization: a BMI > 30 kg/m^2^ in adults over 19 years, >+2 standard deviations (SD) compared to the average for ages between 5 and 19 years. Severe obesity was defined as a BMI >35 kg/m^2^, and morbid obesity as a BMI > 40 kg/m^2^. Small for gestational age (SGA) characterized the patients whose birth length and/or birth weight was <−2 SD of the Usher and McLean tables [40].

The other data collected were as follows: biological parameters for monitoring carbohydrate metabolism from birth to 25 years, hormone levels and hepatic and lipid assessment from the last follow-up.

Diabetes was diagnosed based on the following: blood glucose > 11.1 mmol/L at any time or a fasting blood glucose > 7 mmol/L observed twice, and/or blood glucose > 11.1 mmol/L 2 h after ingestion of 75 g of glucose during an oral glucose tolerance test (OGTT), and/or glycated hemoglobin > 6.5%.

Hypertriglyceridemia was defined as plasma levels > 1.5 g/L. Hypertension was defined as systolic blood pressure ≥ 140 mmHg, and/or a diastolic blood pressure ≥ 90 mmHg.

### 2.3. Statistical Analyses

The characteristics of all patients in the study population are described. Continuous variables are presented by their median [interquartile range IQR_25–75_]. Discrete variables are presented as percentages (%). The Mann–Whitney U test was used for continuous variables and the Chi^2^ test for categorical variables.

## 3. Results

### 3.1. Study Population at Inclusion

Table 1 describes the characteristics of the population at inclusion in the study.

Among the 39 patients, 22 (56.4%) were girls. The patients ranged from 13.9 years to 47.7 years, for a median age of 25.6 years [23.7; 31.7]. The median duration of diabetes at study inclusion was 9.9 years [4.3; 12.3]. The diagnosis of PWS was made at a median age of 29 months [3.75; 157] and 59% of patients had a deletion. For two patients, we did not have the genetic subtype because the parents refused the analysis.

Within our population, 85.3% of patients were followed by a pediatric team during childhood, 58.3% had been treated with GH (16.2% of patients were still being treated, including 1 child and 5 adults), and 48.3% had been hospitalized at least once on a psychiatric unit.

### 3.2. BMI Changes in the Population

Figure 1 describes the evolution of BMI as a function of age in our cohort. 

Most patients had severe obesity. From the age of 15, the median BMI was consistently over 30 kg/m^2^. There was a huge increase in the median BMI between the ages of 15 and 20 years, with a gain of about 6 points. BMI at 15 and 20 years was known for 17 out of 39 patients and the median BMI gain was 6.6 [2.1; 11] for these patients between 15 and 20 years.

### 3.3. History of Diabetes

Table 2 describes the characteristics of T2DM in our population.

At diagnosis, the median age was 16.8 years (from 11 to 24 years), fasting blood glucose was 7 mmol/L [6.1; 13.1] and HbA1c was 8.9% [7.2; 12.9]. All patients were obese except one who was overweight (BMI + 2 SD at 14.5 years). The median BMI was 39 kg/m^2^ [34.9; 45.8] for all patients and +5.75 SD [4.6; 7.5] for children and adolescents. T2DM had been diagnosed in at least one of the parents for 34.5% of the patients and parental overweight or obesity for 53.6% of the patients. Patients were taking a median of one [1;2] antidiabetic medication at diagnosis and three [2;3] at the last follow-up. For 70.3% of the patients, treatment with insulin was necessary from the time of the T2DM diagnosis.

Figure 2 shows age at T2DM diagnosis and evolution of fasting blood glucose and HbA1c as a function of age. 

In our cohort, 23 patients (59%) developed T2DM before adulthood (18 years; Figure 2A). The number of abnormal blood glucose levels increased after the median age of diabetes diagnosis (16.8 years; Figure 2B). With management, median fasting glucose levels were maintained around the normal level in the overall population, but inter-patient variability was wide, as were the changes in HbA1c (Figure 2C). The median HbA1c showed a peak at 20 years, whereas 76.3% of the patients already had T2DM. 

### 3.4. Comparison between Patients Treated with GH and Not Treated with GH

Patients treated with GH during childhood were compared to patients not treated with GH. For three of the patients in our cohort, the status regarding GH treatment was unknown. We therefore excluded them from this analysis, which was carried out on 36 patients: 21 treated with GH and 15 untreated. Table 3 describes the comparison between these two groups. 

We found no significant differences between these two groups other than the median age at inclusion and the age at PWS diagnosis.

## 4. Discussion

In this study, we examined the characteristics of patients with PWS who developed T2DM before the age of 25 years. The median age at diagnosis in our cohort was 16.8 years, and 59% of the patients had developed diabetes before the age of 18. All patients were obese except for one child who was overweight, and the median BMI at diagnosis was 39 kg/m^2^ [34.6; 45.8]. Overall, the patients presented significant comorbidities, both metabolic and psychiatric. We found a high frequency of parental history of T2DM, as well as overweight. In most cases, the diabetes was difficult to control, with great inter-individual variability in HbA1c and fasting blood sugar. At study inclusion, 50% of the patients in our cohort were being treated with insulin, after a median duration of diabetes of 9.9 years (4.3 to 12.3 years). In addition, 26 patients (70.3%) received insulin from the time of their T2DM diagnosis. Among them, 10 had started treatment before the age of 18 years, which might be explained by the limited number of authorized antidiabetic drugs for children (i.e., under 18 years). Eight of these children had to continue insulin after they reached adult age (i.e., 18 years), which reflects both the severity of their diabetes and the difficulty of managing these patients with oral antidiabetic drugs alone. This further suggests that diabetes in young patients with PWS is a somewhat atypical T2DM.

According to the literature, the prevalence of T2DM in adults with PWS is 20 to 25% [21,25]. The mean age at diagnosis varies across studies but is about 20 years old. In 2002, Butler et al. found that 25% of the patients in a cohort of 66 patients with PWS had T2DM, with a mean age at diagnosis of 20.5 ± 8.2 years [21]. More recently, a Japanese team and a Korean team both found younger mean ages at diagnosis, respectively, of 15 years and 15.9 years [27,28]. It should be noted, however, that outside of patients with PWS, T2DM is more common in Japanese adolescents than in their Caucasian counterparts [27]. In 2016, Fintini et al. found that 13.5% of the patients in a cohort of 274 adults and children with PWS had T2DM, for a mean age at diagnosis of 17.9 ± 3.9 years [29]. We chose to study the characteristics of T2DM in young patients with PWS (under 25 years), and our study confirmed the early onset of T2DM in that population, with 16.8 years being the median age at diagnosis and nearly 60% being diagnosed before the age of 18. PWS is therefore a pathology that carries a high risk of T2DM in young people.

The factors associated with early-onset T2DM included high BMI and a family history of diabetes and overweight. Fintini et al. found an association between T2DM onset and BMI, age, and the HOMA index [29]. Our results are consistent with these data. We also noted a rapid increase in BMI between the ages of 15 and 20 years. This period, which corresponds to the transition, is often marked by an increase in eating, behavioral disorders and greater difficulty in controlling food intake and weight. By the age of 19, the median BMI of our cohort had risen to more than 40 kg/m^2^ and remained so, with most of these young patients thus presenting with severe and morbid obesity.

We found no association with the sex ratio or a specific genetic subtype (59% of patients presented a deletion), confirming the data in the literature [17,27,29,32].

As SGA, even in isolation, is a metabolic risk factor—particularly for abnormalities in carbohydrate metabolism [41]—we assessed the prevalence of SGA in our cohort. We found that 33.3% of our patients were born SGA, which is comparable to the 30% observed in the general PWS population [8]. These results are in line with those reported by Tsuchiya et al., who found no difference between diabetic and non-diabetic groups for birth weight or gestational age in a cohort of 65 patients with PWS, including 17 with diabetes [27].

The prevalence of both T2DM (34.5%) and overweight (53.6%) was higher among the parents of our patients than what is usually described in the general population. In France, the frequency of T2DM is estimated at about 5% [22], that of overweight at 30.3%, and that of obesity at 17%, with men, women and children combined [42]. We found no published data regarding family antecedents in the various studies of diabetes in PWS. Our results nevertheless suggest that a family history of T2DM warrants close monitoring to detect the early onset of T2DM in this population—particularly during the transition period.

The prevalence of endocrine comorbidities was comparable to that of the general PWS population [8,9]. However, we found more psychiatric and metabolic comorbidities. Indeed, 47.4% of the patients were being treated with antipsychotic drugs at the time of study inclusion. Laurier et al. studied 154 patients with PWS with a median age of 27 years (from 16 to 54) and found that 39% were treated with antipsychotics [17]. In addition, at inclusion, we noted that 48.3% of our patients had been hospitalized at least once on a psychiatric unit. In general, patients with severe behavioral problems or psychotic disorders are more difficult to manage in terms of diet, resulting in poor weight control. Moreover, the antipsychotic treatments often prescribed in these cases may worsen the weight gain and metabolic syndrome and contribute to the development of T2DM. Likewise, dyslipidemia was more common in our cohort (56% hypertriglyceridemia, almost 60% lower HDLc) than what is usually described in PWS (20–30%) [17,43]. This might also be explained by the prevalence of obesity in our population.

We assessed the implications of GH treatment and compared two groups of patients: those who had been treated with GH and those who had not. The role of GH treatment in the development of diabetes in patients with PWS is controversial: while it can lead to insulin resistance, GH generally has a protective effect on glucose metabolism. Several studies have found a significantly lower BMI in GH-treated patients compared to untreated patients [44,45]. We showed in a previous study that the benefits of GH treatment for BMI, body composition, HbA1c levels, insulinemia and HOMA-R persisted for up to 7 years after treatment cessation [15]. Yet, an Italian study of 274 patients found no difference in the prevalence of carbohydrate metabolism disorders between GH-treated and untreated groups [29]. We had the same observation in our cohort, with no significant difference between the GH-treated and untreated groups apart from the median age and age at diagnosis of PWS. We have no obvious explanation for this. However, almost all our patients were obese, often with severe or morbid obesity that may itself induce T2DM, especially when there is a family history of T2DM and/or overweight and severe behavioral problems.

Glycemic control is more difficult to achieve in patients with PWS and diabetes than in patients with diabetes but no PWS. Metformin treatment, which was prescribed for 94.7% of patients, was often insufficient as a monotherapy. Insulin was or had been used in 70.3% of the cases and the patients had a median of three anti-diabetic treatments [2; 3]. SGLT-2 inhibitors show promise as they allow for significant weight loss along with improved blood sugar control. Three cases of an SGLT-2 inhibitor prescribed in combination with a GLP-1 agonist have been described in PWS, with marked improvement in HbA1c and significant weight loss following initiation of treatment [37,38,46]. However, attention must be paid to the potential side effects (glycosuria favoring urinary tract infections, risk of dehydration) and a case of severe ketoacidosis was described in a 32-year-old patient treated with SGLT-2 inhibitors and a low-carbohydrate diet [47]. It is therefore important to respect the contraindications and follow the patients regularly.

The period of transition from pediatric to adult follow-up carries particular risks. In our previous study conducted from both somatic and psychiatric perspectives, the importance of an organized transition from adapted pediatric care to follow-up by a multidisciplinary adult team was clearly demonstrated: of the 95 adult patients, the 31 who had received appropriate transitional care had better metabolic status (BMI, body fat, metabolic parameters) and less antidepressant treatment [48]. In the present study, we found a peak in median HbA1c at an age of 20 years, whereas the mean age of transition was 19 years. This points to a risk of worsening T2DM and demonstrates the importance of close monitoring of carbohydrate metabolism during the transition as well as comprehensive patient support.

### Limitations and Strengths of the Study 

This study was retrospective, and the size of the cohort was small. We encountered difficulties in finding the parameters for biological and anthropometric monitoring at diagnosis and during the early follow-up of diabetes for the oldest patients. In addition, in three patients, who were among the oldest, the GH status was unknown because there was no mention of GH treatment in the medical records. We further found that screening for abnormalities in carbohydrate metabolism, which is routine in adults, was less common in the children with PWS. Thus, due to missing data, particularly regarding insulinemia, we were unable to conclude on the laboratory parameters for screening for insulin resistance (HOMA-IR). The strength of this study lies in the homogeneous, regular and well-supervised follow-up for Prader–Willi syndrome within the framework of the Reference Center. We were also able to collect family data.

## 5. Conclusions

In our cohort of patients who developed diabetes early in life, before the age of 25 years, we found that the mean age at diagnosis was 16.8 years and that nearly 60% of the patients had developed diabetes before 18 years.

The patients with PWS who are at risk of developing early T2DM are those who are severely obese—especially those undergoing rapid and uncontrolled weight gain in adolescence, and this is independent of whether or not they have been treated with GH. Adolescents with obesity and severe comorbidities, both psychiatric and metabolic, and/or a parental history of overweight or T2DM, should be monitored closely for glucose metabolism, particularly when their BMI increases rapidly. In view of our results, it is essential to do everything possible to avoid excessive weight gain, especially during adolescence, and to detect diabetes early in patients at risk.

## Figures and Tables

**Figure 1 jcm-10-05310-f001:**
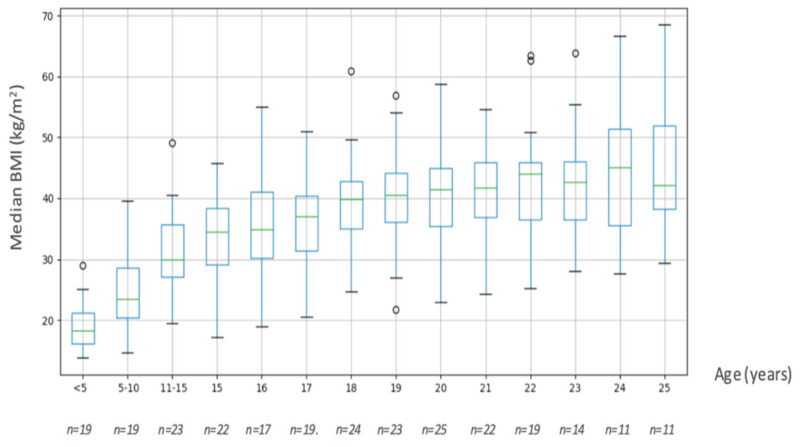
Change in BMI as a function of age in the whole population. BMI was collected between 0 to 5 years, 5 to 10 years, 10 to 15 years, and thereafter from 16 to 20 years.

**Figure 2 jcm-10-05310-f002:**
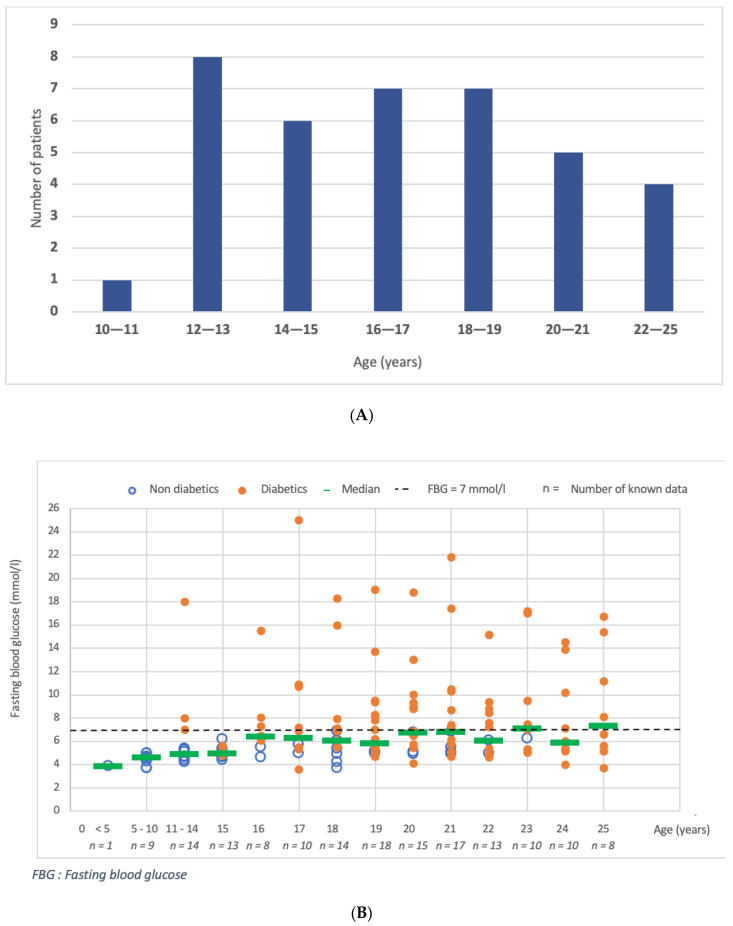
Age at diagnosis and evolution in glucose parameters as a function of age in the population. (**A**): Repartition of patients according to age at diabetes diagnosis (*n* = 38). (**B**): Fasting blood glucose (mmol/L) according to age. The open circle represents plasma glucose levels before diagnosis of T2DM and the full circle represents glucose levels after diagnosis of T2DM. The horizontal line represents the median value. The dotted line represents the cut-off value of 7 mmol/L. *n* below the x axis represents the number of values available at each age. (**C**): HbA1c (%) according to age. The open circle represents HbA1c values before diagnosis of T2DM and the full circle represents HbA1c values after diagnosis of T2DM. The horizontal line represents the median value. The dotted line represents the cut-off value of 6.5%. *n* below the x axis represents the number of values available at each age.

**Table 1 jcm-10-05310-t001:** Description of the population at inclusion (*n* = 39).

	Overall Population *n* = 39 (%)	*n* = Patients Whose Data Was Known
Sex		
Female	22 (56.4)	39
Male	17 (43.6)	39
Age (yr) median [IQR_25–75_]	25.6 [23.1; 31.7]	39
Adults (age ≥ 18 yr)	36 (92.3)	39
Children (age < 18 yr)	3 (7.7)	39
Genetic Subtype		
Deletion	23 (59)	39
Disomy	13 (33.3)	39
Translocation 15; 17	1 (2.3)	39
Abnormal methylation without further precision	2 (5.1)	39
Neonatal Data		
Prematurity (birth < 37 GW)	7 (25)	28
Small for gestational age	9 (33.3)	27
Age (months) at PWS—median [IQR_25–75_]	29 [3,8; 157]	32
Followed by a Pediatric Team	29 (85.3)	34
Duration of Diabetes (yr) median [IQR_25–75_]	9.9 [4.3; 12.3]	38
Endocrine Comorbidities		
GH treatment throughout life	21 (58.3)	36
Thyrotropic insufficiency	16 (42.1)	38
Corticotropic insufficiency	2 (5.6)	36
Hypogonadism	35 (94.6)	37
Metabolic Comorbidities		
Hypertension treated or having been treated	18 (56.3)	32
Hypertriglyceridemia (TG > 1.5 g/L)	22 (56.4)	39
HDL cholesterol < 0.4 g/L	22 (59.4)	37
Psychiatric Comorbidities		
Psychiatric hospitalization	14 (48.3)	29
Medications		
Number of medications per patient—median [IQR_25–75_]	6 [4.5; 7]	39
Diabetes medications		
Metformin	29 (82.9)	35
GLP-1 agonists	17 (51.5)	33
Insulin	19 (50)	38
Sulfonamides	7 (20.6)	34
DPP-4 inhibitors	3 (8.8)	32
Meglitinides	3 (8.8)	34
SGLT-2 inhibitors	6 (16.7)	38
Endocrine medications		
Growth hormone	6 (16.2)	37
Levothyroxine	13 (35.1)	37
Hydrocortisone	2 (5.6)	36
Testosterone for boys	11 (64.7)	17
Estroprogestatives for girls	12 (54.6)	22
Psychotropic medications	20 (52.6)	38
Antipsychotic drugs of all generations	18 (47.4)	38
1st-generation antipsychotic drugs	9 (24.3)	37
2nd-generation antipsychotic drugs	15 (38.5)	39
Benzodiazepines	5 (13.9)	36
Selective serotonin reuptake inhibitors	12 (32.4)	37

Data are expressed as Number (%) or Median [IQR], yr: years, GW: weeks of gestation, PWS: Prader-Willi syndrome, GH: growth hormone, TG: triglycerides, HDL: high-density lipoproteins, DPP4: dipeptidyl peptidase-4, GLP-1: glucose-like peptide 1, SGLT-2: sodium/glucose cotransporter 2.

**Table 2 jcm-10-05310-t002:** Description of diabetes in the population (*n* = 39).

	Overall Population *n* = 39 (%)	*n* = Patients for Whom the Data Were Known
History of Diabetes		
Age (yr) at T2DM—median [IQR_25–75_]	16.8 [14.1; 19]	38
Diagnostic criterion: FBG >7 mmol/L	13 (68.4)	19
Diagnostic criterion: HbA1c > 6.5%	22 (95.6)	23
Diagnostic criterion: pathological OGTT	1 (4.6)	22
Polyuropolydipsic syndrome	8 (42.1)	19
HbA1c at diagnosis—median [IQR_25–75_]%	8.9 [7.2; 12.9]	23
FBG at diagnosis mmol/L- median [IQR_25–75_]	7 [6.1; 13.1]	19
BMI at diagnosis—median [IQR_25–75_] kg/m^2^	39 [34.9; 45.8]	29
Family Antecedents		
Parental antecedents of T2DM	10 (34.5)	29
Parental obesity or overweight	15 (53.6)	28
Medications Taken at Diabetes Diagnosis		
Number of medications per patient at diagnosis	1.5 (0.2)	39
Endocrine medication		
Growth hormone	10 (58.8)	17
Levothyroxin	5 (16.1)	31
Hydrocortisone	2 (6)	33
Testosterone in boys	4 (28.6)	14
Estroprogestatives in girls	8 (50)	16
Psychotropic medications	7 (25.8)	28
Antipsychotic drugs—all generations	8 (26.7)	30
1st-generation antipsychotic drugs	1 (3.6)	28
2nd-generation antipsychotic drugs	8 (25.8)	31
Selective serotonin reuptake inhibitors	2 (6.3)	32
History of Diabetes Treatment		
Number of medications per patient at diagnosis	1 (1; 2)	27
Number of medications per patient at inclusion	3 (2; 3)	37
Metformin	36 (94.7)	38
GLP-1 agonists	27 (71)	38
Insulin	26 (70.3)	38
Sulfonamides	17 (51.5)	33
DPP-4 inhibitors	11 (35.5)	31
Meglitinides	9 (30)	30
SGLT-2 inhibitors	6 (19.4)	31

Data are expressed as Number (%) or Median [IQR], yr: years, T2DM: type 2 diabetes mellitus, FBG: fasting blood glucose, HbA1c: glycated hemoglobin, OGTT: oral glucose tolerance test, BMI: body mass index, DPP4: dipeptidyl peptidase-4, GLP-1: glucose-like peptide 1, SGLT-2: sodium/glucose co-transporter 2.

**Table 3 jcm-10-05310-t003:** Comparison of patients previously treated with GH with those never treated (*n* = 36).

	Treated with GH *n* = 21 (58.3%)	Not Treated with GH *n* = 15 (41.7%)	*p*-Value
General Characteristics			
Sex ratio boys/girls	10/11 (47.6/52.4)	7/8 (46.7/53.3)	NS
Age (yr) median [IQR_25–75_]	23.4 [20.4; 25.6]	30.3 [25.8; 33]	**0.0028**
Genetic Subtype			NS
Deletion	13 (61.9)	8 (53.3)	
Disomy	5 (23.8)	7 (46.7)	
Translocation 15; 17	1 (4.8)	0 (0)	
Methylation anomaly without precision	2 (9.5)	0 (0)	
Neonatal Data			
Prematurity (birth < 37 GA)	5 (29.4)	2 (18.2)	NS
Intrauterine growth retardation	7 (46.7)	6 (50)	NS
Age (m) at PWS diagnosis—median [IQR_25–75_]	6.5 [1.2; 29.5]	162 [73.5; 202]	**0.0002**
Followed by a Pediatric Team	17 (94.4)	11 (73.3)	NS
History of Diabetes			NS
Age (yr) at T2DM—median [IQR_25–75_]	16.8 [13.8; 20]	16 [14; 18.5]	
BMI at diagnosis—median [IQR_25–75_]	39 [34.9; 45.4]	39.1 [35.7; 45.8]	
HbA1c at diagnosis—median [IQR_25–75_]	8.9 [7.3; 12.4]	9 [7.1; 12.6]	
Fasting blood glucose at diagnosis—median [IQR_25–75_]	7.7 [6.4; 12.1]	10 [7.6; 14.3]	
Diagnostic criterion: FBG > 7 mmol/L	8 (61.5)	5 (83.3)	
Diagnostic criterion: HbA1c > 6.5%	16 (100)	6 (85.7)	
Diagnostic criterion: pathological OGTT	0 (0)	1 (14.3)	
Family Antecedents			NS
Parental antecedents of T2DM	4 (28.6)	5 (55.6)	
Parental antecedents of obesity or overweight	8 (72.7)	4 (44.4)	
Metabolic Comorbidities			NS
Hypertension treated or having been treated	9 (52.9)	7 (53.9)	
Hypertriglyceridemia (TG > 1.5 g/L)	11 (55)	8 (53.3)	
HDL cholesterol < 0.4 g/L	13 (68.4)	7 (46.7)	
Psychiatric Comorbidities			NS
Psychiatric hospitalization	8 (47)	5 (45.6)	
Medications at Inclusion			NS
Current number of medications per patient—median [IQR_25–75_]	6 [4.8; 7.3]	6 [3; 7]	
Diabetes medications			NS
Number of medications per patient at diagnosis—median [IQR_25–75_]	1 [1; 2]	1 [1; 1.5]	
Number of medications per patient at inclusion	3 [1; 3]	2 [2; 3]	
Metformin	14 (77.8)	11 (84.6)	
GLP-1 agonists	9 (50)	6 (46.2)	
Insulin	11 (55)	6 (40)	
Sulfonamides	3 (17.7)	3 (21.4)	
DPP-4 inhibitors	4 (23.6)	2 (14.3)	
Meglitinides	2 (12.5)	1 (6.7)	
SGLT-2 inhibitors	4 (25)	2 (13.3)	
Endocrine medication			NS
Growth hormone	5 (25)	0 (0)	
Testosterone in boys	7 (70)	5 (71.4)	
Estroprogestatives in girls	5 (50)	5 (62.5)	
Psychotropic medication	12 (66.7)	8 (53.3)	NS

Data are expressed as Number (%) or Median [IQR].

## Data Availability

The data presented in this study are available on request from the corresponding author.

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
