# Peer review of "Diabetes Mellitus in Prader-Willi Syndrome: Natural History during the Transition from Childhood to Adulthood in a Cohort of 39 Patients"

_jcm, 2021, doi:10.3390/jcm10225310_

Round 1

Reviewer 1 Report

Dear Authors,

first of all i would like to thank you for this paper. Is really well written, it is clear and easy to read. 

Some phrases need to be changed in order to be more flowing. So i suggest a minor revision of the english text which looks still good.

In order to improve the paper you could please better explain the limitations and enrich the manuscript with this citation:

Hypogonadism in Patients with Prader Willi Syndrome: A Narrative Review.

Napolitano L, Barone B, Morra S, Celentano G, La Rocca R, Capece M, Morgera V, Turco C, Caputo VF, Spena G, Romano L, De Luca L, Califano G, Collà Ruvolo C, Mangiapia F, Mirone V, Longo N, Creta M.Int J Mol Sci. 2021 Feb 17;22(4):1993. doi: 10.3390/ijms22041993.

Reviewer 2 Report

Prader-Willi syndrome is rare and knowledges about is are always interesting. This study describe diabetes mellitus in 39 patients with PWS and followed  in big centers of reference.

Some remark nevertheless

In the abstract

- for three patients GH status was unknown which is surprising. Do we know more about theses patients ? are they lost of follow up ?

- percent of overweight (53.6%) is written with comma and not a point

In the results

  • table 1 : i don't see the interest to have the numbers of all medications and diabetes medications. What is the difference between inclusion and last follow up ?
  • table 2 : i would like to know more about diabetes in PWS patients : what was HbA1C at last follow up ? did the patient have retinopathy or nephropathy ? how were diabetes medications associated ? what was the effect of GLP-1 agonist on weight and HbA1C in PWS patients ?
  • page 7, there is difference in results in the text and in table 2 (HbA1C  9.1% and 8.9 : what is the correct one ? FBG 7.7 mmol and 7.0 mmol and range of BMI is also different).
  • table 3 : what is the interest of number of medications at inclusion ?

In the discussion

- as the authors are in reference centers, do they have date comparing percent of diabetes patients with or withou GH treatment ? It appears that GH seems to be safe on diabetes for theses patients and that GH treatment improves weight at adulthood. Do patients treated with GH have diabete less often than patients without GH treatment ?
